# CRISPR-TAPE: protein-centric CRISPR guide design for targeted proteome engineering

Daniel Paolo Anderson[1,†] ID, Henry James Benns[1,2,†] ID, Edward William Tate[2] ID & Matthew Andrew Child[1,*] ID

## Abstract

**Rational molecular engineering of proteins with CRISPR-based approaches is challenged by the gene-centric nature of gRNA design tools. To address this, we have developed CRISPR-TAPE, a protein-centric gRNA design algorithm that allows users to target specific residues, or amino acid types within proteins. gRNA outputs can be customized to support maximal efficacy of homology-directed repair for engineering purposes, removing time-consuming *post hoc* curation, simplifying gRNA outputs and reducing CPU times.**

**Keywords** CRISPR; genome engineering; protein-centric; target prioritization
**Subject Categories** Biotechnology & Synthetic Biology; Computational Biology
**Mol Syst Biol. (2020) 16: e9475**

## Introduction

Functional genomics has been revolutionized by the discovery of the CRISPR prokaryotic defence system (Barrangou *et al*, 2007; Jinek *et al*, 2012) and its conversion into an effective and efficient mechanism for genome engineering (Cong *et al*, 2013; Mali *et al*, 2013; Ran *et al*, 2013). CRISPR technologies depend on the targeting of an RNA-guided endonuclease to a defined sequence location within the genome. This system has been harnessed for a variety of genome modification strategies including gene knockouts (through incorrect repair of breaks) and site-directed mutagenesis (through increased efficiency of homology-directed repair incorporating DNA templates at genomic regions close to the breaks). In most applications, accurate targeting of the nuclease to the genomic locus takes priority over the specific nucleotide position of enzymatic activity. While engineering of the nuclease has driven diversification of the technologies that this system can support (Pickar-Oliver & Gersbach, 2019), the molecular rules governing nuclease targeting remain the same; the genome address is encoded within a guide RNA sequence (gRNA), defined as a 20-nucleotide stretch of genomic DNA preceding a protospacer adjacent motif (PAM). An abundance of bioinformatic tools is available for the identification and selection of unique CRISPR gRNA sequences required for nuclease targeting of individual genes (Wilson *et al*, 2018; Thomas *et al*, 2019), but search algorithms have remained gene-centric (Fig 1A). This has limited the wider exploitation of these technologies by protein- and proteome-engineers, and researchers seeking to modify specific amino acids or protein sequences such as catalytic residues within enzyme active sites.

At present, gRNAs targeting protein-coding regions within a genomic locus are anonymous within total gRNA outputs from existing design tools that non-specifically target the entire input region of DNA (Table EV1). These gRNA lists subsequently require extensive time-consuming manual curation to identify those targeting specific protein regions of interest. The principle challenge encountered by existing design algorithms is the non-linear correlation of genomic sequence (intronic and exonic) with protein coding sequence (spliced exonic). This often requires users to manually distinguish between intron–exon sequences and pair target amino acid/s with proximal gRNAs. The gene-centric focus of existing algorithms has inadvertently led to an absence of protein design considerations, such as the distance of the nuclease cut site from a protein feature of interest or mutation site. This is important as the increased efficiency of homology-directed repair (HDR) at double-strand breaks driven by the activity of the nuclease has a limited range. The efficiency of HDR decreases with increasing distance from the nuclease cut site, up to a maximum range of 30 nt (Paquet *et al*, 2016). If the distance of the mutation site of interest from the nuclease cut site is > 30 nt, there is no increase in the efficiency of HDR afforded by the use of the RNA-guided nuclease. This emphasizes the need to account for this when selecting gRNAs for directed mutagenesis strategies, where optimal HDR is essential. Existing tools are unable to directly query gRNAs targeting specific amino acids or positions of interest such as sites of post-translational modification (PTM).

## Results and Discussion

To address this, we have developed CRISPR-TAPE, a protein-centric CRISPR gRNA design algorithm for Targeted Proteome Engineering (Fig 1B and computational methods). CRISPR-TAPE is run as a

---

1 Department of Life Sciences, Imperial College London, London, UK
2 Department of Chemistry, Imperial College London, London, UK
*Corresponding author. Tel: +44 (0) 207 594 5402; E-mail: m.child@imperial.ac.uk
†These authors contributed equally to this work

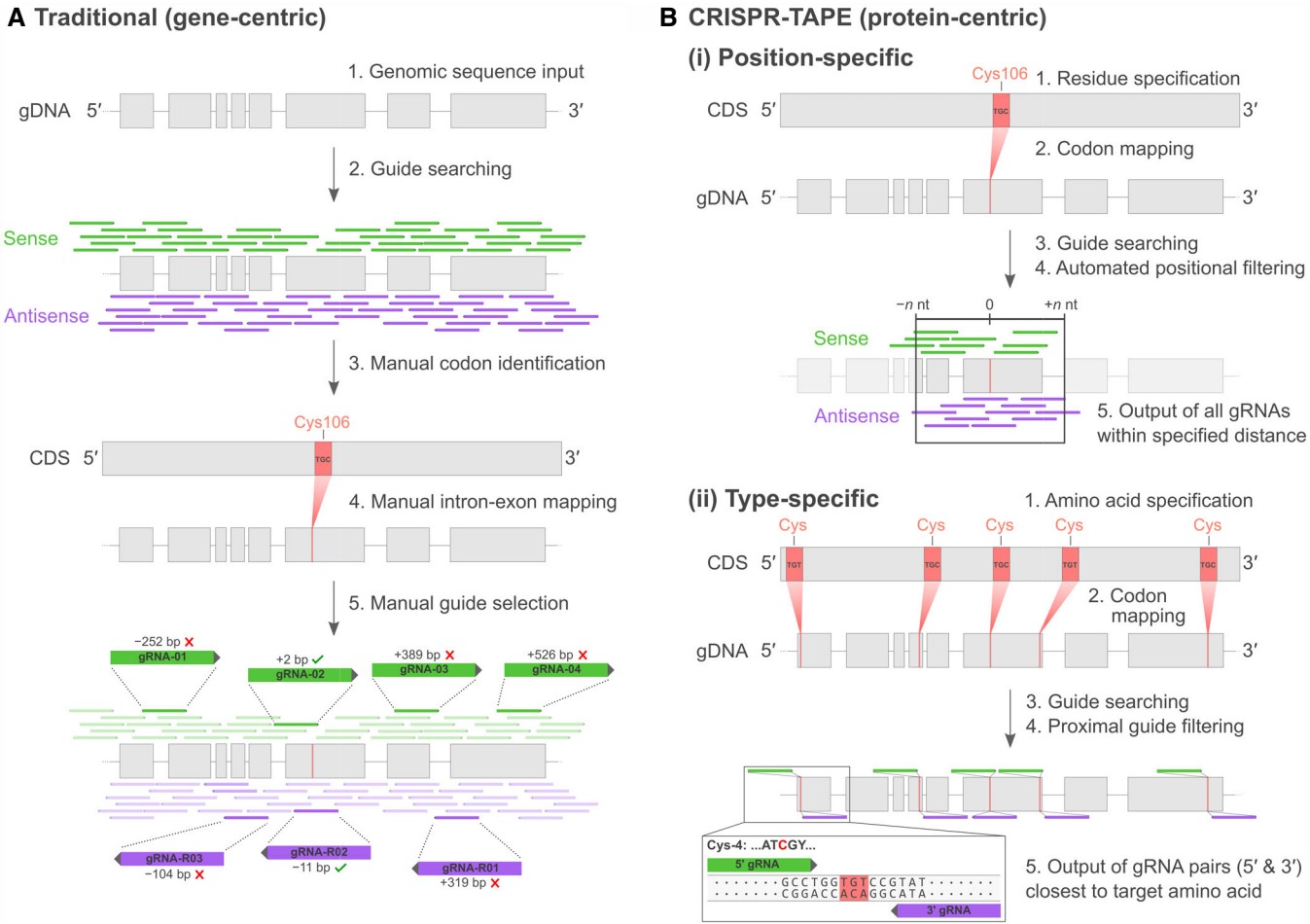

**Figure 1. Automated protein-centric identification of CRISPR gRNAs using CRISPR-TAPE.**

A   Traditional gene-centric approach for amino acid-targeted CRISPR gRNA design. Users input the genomic DNA (gDNA) locus of their target protein into an existing gene-centric gRNA design tool (1). Gene-centric algorithms typically query chosen PAM sequences against the entire genomic locus without subsequent positional filtering, resulting in a complex output of large numbers of gRNAs (2, and Fig EV2). The codon positions of the target amino acid(s) are then manually identified in the protein coding sequence (CDS; 3) and mapped onto its associated gDNA sequence, for users to be able to account for complex intron–exon gene structures (4). From the extensive list of gRNAs generated in 2, users must initiate time-consuming manual curation to identify gRNAs with a nuclease cut site in close proximity to residues on both sense and antisense gDNA strands (5).

B   Workflow schematics of position-specific (i) and type-specific (ii) functions of CRISPR-TAPE. Following selection of a specific residue position (i) or amino acid type (ii; 1), the corresponding codon(s) from the target protein's coding sequence (CDS) are mapped onto its genomic DNA (gDNA) locus (2). A selected PAM sequence is then queried against the gDNA to identify guides (3) and filtered according to a user-defined cut site distance from the specified residue position (i) or by the two most proximal gRNAs that target up- and downstream of each protein sequence representation of an amino acid type (ii; 4).

freely available stand-alone python script and custom executable application (www.laboratorychild.com/crispr-tape) (Fig EV1 and Appendix). The CRISPR-TAPE algorithm is based upon a multidimensional matrix mapping strategy, where the coding sequence is split into codons and these translated to a protein sequence (Fig EV2). The corresponding codon positions relative to the genomic sequence are stored for each residue. This numerical indexing enables the algorithm to account for complex intron–exon gene structures and supports protein-focused gRNA searches. Users query amino acid types (e.g. cysteines) or individual amino acids at specific positions of interest within their input protein sequence (e.g. cysteine 106) facilitating direct interrogation of protein domains, catalytic motifs and PTMs. To our knowledge, these features are unique to CRISPR-TAPE. The reduced complexity of the search

space greatly reduces computational burden and central processing unit (CPU) times for gRNA identification; applied to the ~ 80 Mb genome of the model eukaryotic pathogen *Toxoplasma gondii*, for a single specified amino acid within an input protein coding sequence, guides directing Cas9 to within 30 nucleotides of the residue for optimal HDR are typically output in < 5-s (Fig EV3A). Although comparisons with existing tools are inherently unfair (due to fundamental differences in source codes), identical searches using existing gene-centric tools require a processing time in the order of minutes. For a better comparison, we benchmarked CRISPR-TAPE performance against our source code; we deployed our code for non-directed guide identification within genomic loci (i.e. equivalent to traditional total guide output for genomic loci). Compared with our own source code deployed for total guide identification, CPU

times required by CRISPR-TAPE for searches are greatly reduced. gRNAs for position-specific queries are processed at an average rate of ~ 330 nucleotides per second (Fig EV3A). For type-specific queries, using leucine as a linearly distributed amino acid in our test gene set (Fig EV3B), gRNAs are processed at an average rate of ~ 27 nucleotides per second (Fig EV3A). These rates are approximately 40- and 3.2-fold faster than basic gRNA identification for the position-specific and type-specific search modes, respectively.

CRISPR-TAPE reduces output gRNA complexity as guide sequences are automatically curated. gRNA outputs are provided in relation to the specified amino acid or amino acid type within the genomic locus and distributed according to distance of the nuclease cut site from the specified amino acid/s to support efficient HDR strategies (Fig 2). gRNAs therefore require no further manual curation to correlate genome targeting with the protein coding sequence (unlike traditional gene-centric gRNA outputs, Table EV1). While it is challenging to quantify timesaving performance improvements when comparing automated to manual curation (as it involves highly variable parameters such as user experience and background knowledge), the benefit of automated curation of gRNAs when using CRISPR-TAPE is expected to be substantial. Distinct from gene-centric search tools, CRISPR-TAPE prioritizes proximity of the nuclease cut site to a specified amino acid type/position over all other guide selection and scoring criteria.

CRISPR-TAPE performance was then assessed for a potential genome-scale protein engineering application; we identified a well-characterized dataset for the malaria parasite *Plasmodium falciparum*, where a known PTM (*S*-palmitoylation) has been globally profiled but site-specific information regarding the modified amino acid (cysteine) is absent (Jones *et al*, 2012). Such a dataset is suitable for the CRISPR-TAPE "type-specific" function, where pairs of gRNAs proximal to a user-defined amino acid type are output simultaneously (Fig 2B). We focused on 55 putatively palmitoylated proteins the authors of this work identified as being representative

of common and uncommon palmitoyl protein classes (Jones *et al*, 2012). We first used a traditional gene-centric search algorithm to identify all gRNAs present within the input gene sequences (Peng & Tarleton, 2015). CPU time required for the initial gRNA search was on average 25 s per gene. In our experience, manual curation time required to identify pairs of gRNAs targeting individual amino acids takes a minimum of 5 min/residue. Considering that the average number of cysteines/gene in this dataset is 11 and including the average CPU time, this can be extrapolated to ~ 55 min per gene. We then applied CRISPR-TAPE to the same dataset, using the "type-specific" option to identify pairs of gRNAs in proximal positions to all potential sites of *S*-palmitoylation (i.e. all cysteine residues). The average processing time for each protein was 6-s, with no downstream curation of gRNA outputs required. Applied in this way and compared with a traditional gene-centric gRNA design tool, CRISPR-TAPE was at least 300 times faster per cysteine being targeted. These data are summarized in Table EV2.

The profiling of amino acid reactivity within proteins is a rapidly expanding field, with this chemical proteomic strategy having been successfully applied to profile the reactivity of serine (Kidd *et al*, 2001), cysteine (Weerapana *et al*, 2010), lysine (Hacker *et al*, 2017), histidine (Jia *et al*, 2019), tyrosine (Hahm *et al*, 2020) and methionine residues (Ohata *et al*, 2020). A method to efficiently prioritize these reactive sites according to their contribution to protein function would optimize pipelines for target-based screening platforms. Such a method could be supported by CRISPR-based site-directed mutagenesis via homology-directed repair. This would support CRISPR base editing using Cas9-cytidine/adenine deaminase fusions where particular amino acid switches are enzymatically inaccessible, such as threonine to serine. CRISPR-TAPE is focused on protein-led gRNA design, with guide outputs being for suitable for all CRISPR systems (e.g. base editors (Rees & Liu, 2018)). To test the performance of our algorithm applied to specific sites of interest, we selected a subset of 12 ligandable sites on 10 proteins from a recent reactive lysine

**A**

| Distance from Amino Acid (bp) | gRNA Sequence | PAM | Strand | G/C Content (%) | Off Target Count | Notes |
|---|---|---|---|---|---|---|
| -25 | AGATGGCCTGGTTCACTCTG | CGG | reverse | 55 | 0 | No leading G. |
| -20 | TCCCCTCCGCAGAGTGAACC | AGG | forward | 65 | 0 | No leading G. |
| -13 | TGCAGATGAGGTAGATGGCC | TGG | reverse | 55 | 0 | No leading G. |
| -8 | GCCGGTGCAGATGAGGTAGA | TGG | reverse | 60 | 0 | |
| -1 | ACCTTGCGCCGGTGCAGATG | AGG | reverse | 65 | 0 | No leading G. |
| C-143 | | | | | | |
| -1 | GCCATCTACCTCATCTGCAC | CGG | forward | 55 | 0 | |
| 6 | ACCTCATCTGCACCGGCGCA | AGG | forward | 65 | 0 | No leading G. |
| 7 | CAGAGGGACTGACCTTGCGC | CGG | reverse | 65 | 0 | No leading G. |
| 23 | GCTGTACAGTTTCTTTCAGA | GGG | reverse | 40 | 0 | |
| 24 | AGCTGTACAGTTTCTTTCAG | AGG | reverse | 40 | 0 | No leading G. |

**B**

| Amino Acid Position | Adjacent amino acids | gRNA Sequence 5' of Amino Acid | PAM | gRNA Strand | gRNA G/C Content (%) | Distance of Cut Site from Amino Acid (bp) | Notes | gRNA Off Target Count |
|---|---|---|---|---|---|---|---|---|
| 34 | KAC*VF | CTACATCGCGGTCAAGGACA | AGG | forward | 55 | -8 | No leading G. | 0 |
| 54 | AMC*PI | ACGCTCGACAATGGGGCACA | TGG | reverse | 60 | 1 | No leading G. | 0 |
| 143 | LIC*TG | ACCTTGCGCCGGTGCAGATG | AGG | reverse | 65 | -1 | No leading G. | 0 |
| 160 | AEC*LA | GATCTCGTCCGCAAGGCACT | CGG | reverse | 60 | 1 | | 0 |
| 168 | MNC*AK | AGTTCATGATCTCGTCCGCA | AGG | reverse | 50 | -16 | No leading G. | 0 |

| | | gRNA Sequence 3' of Amino Acid | PAM | gRNA Strand | gRNA G/C Content (%) | Distance of Cut Site from Amino Acid (bp) | Notes | gRNA Off Target Count |
|---|---|---|---|---|---|---|---|---|
| | | AGTGTGCGGCACGAAGACGC | AGG | reverse | 65 | 1 | No leading G. | 0 |
| | | TGACGAGACGCTCGACAATG | GGG | reverse | 55 | 5 | No leading G. | 0 |
| | | GCCATCTACCTCATCTGCAC | CGG | forward | 55 | -1 | | 0 |
| | | GACCATCGCCGAGTGCCTTG | CGG | forward | 65 | 1 | | 0 |
| | | CGAGATCATGAACTGCGCCA | AGG | forward | 55 | 1 | No leading G. | 0 |

**Figure 2. Auto-curated gRNA outputs provided by CRISPR-TAPE for targeting specific amino acid positions or types.**

Output files generated by CRISPR-TAPE using the latest *Toxoplasma gondii* GT1 genome release (ToxoDB-46), and querying *Toxoplasma* gene ID TGGT1_242330 with (A) the "Position-specific" function, identifying gRNAs within 30 nucleotides of a target cysteine at position 143, and (B) the "Type-specific" function, with gRNA pairs provided for all cysteines present in the CDS.

profiling dataset within the human cancer cell proteomes of MDA-MB-231, Ramos and Jurkat cells (Hacker *et al*, 2017). We focused on kinases as targets of proven therapeutic value, and sought to identify a panel of gRNAs that would direct Cas9 to within 30 nucleotides of a target lysine (to support efficient HDR for mutational interrogation of its function). CPU times for CRISPR-TAPE were on average double that of the online gene-centric tool CHOPCHOP (Labun *et al*, 2019). This difference could be due to a number of factors including the processing power of the consortia-funded data servers that support these community tools. Moreover, user-dependent factors such as memory/power consumption, and latency times between user input and programme execution make direct comparison of processing speeds for different algorithms inherently unequal. Fundamentally, the improvement in computation processing speed is a reflection of the restricted search space that results from targeting specific amino acids or amino acid types, and not the entire gene locus. On average, CRISPR-TAPE identified 10 gRNAs/reactive lysine within the defined 30 nt window upstream and downstream of the residue codon, with these gRNA panels auto-curated and immediately ready for downstream applications. A conservative estimation for the manual

curation of tightly targeted gRNA panels from dense gRNA output arrays provided by gene-centric tools would be a minimum of 5 min/gRNA. For the identical panel of 10 gRNAs/reactive lysine, this can be extrapolated to an expected total manual curation time of 50 min/lysine. Accounting for differences in raw processing speed, this makes CRISPR-TAPE at least 10 times faster than gene-centric tools used in this way, and these data are summarized in Table EV3.

CRISPR-TAPE source code is freely available (github.com/LaboratoryChild/CRISPR-TAPE) and organism-adaptable and includes standard guide-design features such as off-target scoring and identification of guide sequences predicted to be ineffective (Ran *et al*, 2013; Haeussler *et al*, 2016; Wilson *et al*, 2018). The code can also be expanded to include amino acid motif-based proteome engineering strategies, batch processing and next-generation genome editing tools such as Prime Editing (Anzalone *et al*, 2019). We anticipate that CRISPR-TAPE will support existing gene-centric tools and empower the proteome engineering community. It will accelerate the application of CRISPR-based methods for targeted protein modification, *in vivo* protein evolution and amino acid prioritization in drug discovery.

# Materials and Methods

**Reagents and Tools table**

| Reagent/resource | Reference or source |
|---|---|
| **Experimental models** | |
| 3D7 (*P. falciparum*) | Jones *et al* (2012) |
| MDA-MB-231, Ramos and Jurkat cancer cell lines (*Homo sapiens*) | Hacker *et al* (2017) |
| **Software** | |
| Graphpad Prism software v8 | https://www.graphpad.com/scientific-software/prism/ |
| EuPaGDT | http://grna.ctegd.uga.edu (Peng & Tarleton, 2015) |
| CHOPCHOP v3 | https://chopchop.cbu.uib.no (Labun *et al*, 2019) |
| Python 3.7.3 | https://www.python.org |
| Anaconda 3.7 | https://www.anaconda.com |
| **Other** | |
| Genome/gene sequence data (*T. gondii*) | https://www.toxodb.org/ (Gajria *et al*, 2008) |
| Genome/gene sequence data (*P. falciparum*) | https://www.plasmodb.org/ (Bahl *et al*, 2003) |
| Genome/gene sequence data (*H. sapiens*) | http://www.ensembl.org/ (Cunningham et, 2019) |
| Code Package: Numpy 1.18.0 3 | https://numpy.org |
| Code Package: Pandas 0.25.3 | https://pandas.pydata.org |
| Code Package: tkinter 8.6 | https://tkdocs.com |
| Code Package: Pyinstaller 3.5 | https://www.pyinstaller.org |

**Methods and Protocols**

*User input*
CRISPR-TAPE requires several user inputs to function correctly. Once CRISPR-TAPE has loaded, user inputs are added via entry boxes in the graphical user interface (GUI) window. Both running modes (custom application or raw Python script) require the organism genome to be accessible for successful gRNA off-target site identification. This is done by downloading the

genome of the organism of interest, placing the genome file within the specified location (see below) and then renaming it to "INSERT_ORGANISM_GENOME_HERE.txt" (and replacing the existing file with that name). Genome files can be downloaded in FASTA format (.fa) and directly renamed. The genome.txt file must be located within the same directory as the CRISPR-TAPE executable application (or Python script if using command line) for the programme to recognize and import the genome of interest when running.

Within the GUI window, the user begins by specifying a file-name for the guide RNA (gRNA) output table. Users then specify input the genomic loci sequence of the protein of interest (5′–3′ orientation), with introns and UTRs in lowercase and exons in uppercase. If UTRs are not included, users can also include 100 bases (or longer) upstream and downstream of the genomic loci to enable identification of gRNAs targeting amino acids in close proximity to the protein N- and C-termini, respectively. While this is not strictly required and these input boxes can be left empty, exclusion of flanking sequences may limit the number of gRNAs identified at these sites. CRISPR-TAPE currently supports "NGG", "YG" and "TTTN" PAMs, and these can be selected through a checkbox. The open source availability and modularity of the CRISPR-TAPE source code allows for easy incorporation of additional PAM sequences (see the download-associated README file and Appendix for detailed information). Users then choose to target an amino acid at a specific position by inputting the position of the residue within the protein sequence (OPTION 1), or target all amino acids of a certain type by inputting the amino acid single letter code (OPTION 2). Within OPTION 1, the user may also specify the maximum base distance of guides from the residue of interest to limit the range of outputted gRNAs. If no distance is specified, all guides within the input locus are identified.

### Pre-processing

The pre-processing stage is required to import and store the genome of the organism of interest from the "INSERT_ORGA-NISM_GENOME_HERE.txt" and perform some basic manipulations of the user inputted sequences. The inputted genomic loci and stored genome are reverse-complemented for reverse strand guide identification and off-target counting, respectively. The uppercase bases within the genomic loci are recognized and converted into a coding sequence (CDS). This concatenated exon sequence is then aligned to the user inputted CDS. If these are not identical, gRNAs will not be generated. This launches an error pop-up informing the user that that the intron–exon structure of the gene in relation to the coding sequence does not match and should be checked. The positions of exonic bases relative to the genomic loci are then stored in a 1D array for matrix generation in the next stage. The inputted CDS is translated into an amino acid sequence, and the matrix position of each residue is stored.

### CRISPR-TAPE

#### OPTION 1 and 2 shared processes

A multidimensional matrix is generated to compile information for each amino acid within the input CDS. This matrix consists of the position of each residue in the protein sequence, the specific codon that codes for that amino acid, and the position of each of the corresponding codon bases within the inputted genomic loci. The programme searches for the positions of the user-specified PAM on both the forward and reverse strands of the genomic locus and outputs the position of the base immediately 5′ of the nuclease cut site. This position is later used to determine an initial crude distance value between the cut site and the base 5′ of the codon. This positional information is used to generate forward and reverse strand lists of all potential gRNAs within the inputted genomic loci,

arranged by position. gRNA sequences and associated nuclease cut site positions are converted to a dataframe and G/C percentages of gRNA sequences calculated.

The matrix position of the 5′ or 3′ base in the codon triplet is used to determine which base 5′ of the gRNA(s) cut site position is in closest proximity to the user-specified residue(s) within the genomic loci, and this used to identify the corresponding gRNA sequences 5′ and 3′ to the target. When gRNAs are initially identified, their position is indexed as the base 5′ of the nuclease cut site regardless which strand they are on in relation to the inputted genomic locus. A subsequent function then corrects this distance accounting for the strand; the distance between the 5′ or 3′ base of the nuclease cut site and 5′ or 3′ base of the codon is calculated and appended to the gRNA dataframe(s). "Leading G" and "poly-T" information of gRNA sequences is also determined and appended to the dataframe. Off-targets are searched and counted for each gRNA sequence in the inputted organism genome and its reverse complement. The tool does not provide a score for gRNA outputs.

#### OPTION 1 specific

Guides located beyond the user-specified distance are removed from the dataframe. The gRNA dataframe is then split (based on positional information) into gRNAs 5′ and 3′ of the residue. The dataframe of 5′ gRNAs is arranged by decreasing distance between cut site and amino acid, and the dataframe of 3′ gRNAs arranged by increasing distance.

#### OPTION 2 specific

gRNAs with > 75% G/C content and associated cut site positions are removed from the dataframe.

#### Output

The list of the gRNAs generated by CRISPR-TAPE is outputted within the CRISPR-TAPE home directory to a ".csv" file specified by the user.

The output of OPTION 1 consists of:

1. gRNA Sequence: The gRNA sequence identified by the programme.
2. PAM: The specific PAM immediately adjacent to the gRNA.
3. Strand: The orientation of the DNA strand the gRNA targets relative to the sense of the inputted genomic loci.
4. G/C Content: The percentage of the gRNA sequence consisting of "G" and "C" bases.
5. Distance from aa (bp):

- If positive and "Strand" = forward: gRNA is upstream of the amino acid, the distance is measured from the base on the right-hand side of the nuclease cut site to the base on the left-hand side of the codon (5′–3′).
- If negative and "Strand" = forward: gRNA is downstream of the amino acid, the distance is measured from the base on the left-hand side of the nuclease cut site to the base on the right-hand side of the codon (5′–3′).
- If positive and "Strand" = reverse: gRNA is upstream of the amino acid, the distance is measured from the base on the left-hand side of the nuclease cut site to the base on the left-hand side of the codon (5′–3′)

- If negative and "Strand" = reverse strand: gRNA is downstream of the amino acid, the distance is measured from the base on the right-hand side of the nuclease cut site to the base on the right-hand side of the codon (5′–3′).

6  Notes: Does the gRNA contain a poly-T sequence indicated by a tandem of four or more Ts? Does the gRNA have a leading G at position 1 in the gRNA? Is the G/C content over 75%?

7  Off-target Count: The number of off-target sites the gRNA may target.

The output of OPTION 2 consists of:

1  Amino Acid Position: The position of the amino acid within the amino acid sequence.

2  Adjacent amino acids: The four amino acids immediately surrounding the residue being targeted. The target residue is indicated by "*".

3  5′ gRNA Sequence: The sequence of the gRNA closest in proximity upstream of the amino acid.

4  3′ gRNA Sequence: The sequence of the gRNA closest in proximity downstream of the amino acid.

5  PAM: The specific PAM immediately adjacent to the 5′ or 3′ gRNA.

6  Strand: The orientation of the DNA strand the 5′ or 3′ gRNA targets relative to the sense of the inputted genomic loci.

7  G/C Content: The percentage of the 5′ or 3′ gRNA sequence consisting of "G" and "C" bases.

8  Distance of cut site from Amino Acid (bp):

- If "Strand" = forward and the gRNA is 5′ of the residue, the distance is measured from the base on the right-hand side of the nuclease cut site to the base on the left-hand side of the codon (5′–3′).

- If "Strand" = forward and the gRNA is 3′ of the residue, the distance is measured from the base on the left-hand side of the nuclease cut site to the base on the right-hand side of the codon (5′–3′).

- If "Strand" = reverse and the gRNA is 5′ of the residue, the distance is measured from the base on the left-hand side of the nuclease cut site to the base on the left-hand side of the codon (5′–3′).

- If "Strand" = reverse and the gRNA is 3′ of the residue, the distance is measured from the base on the right-hand side of the nuclease cut site to the base on the right-hand side of the codon (5′–3′).

9  Notes: Does the 5′ or 3′ gRNA contain a poly-T sequence indicated by a tandem of four or more Ts? Does the gRNA have a leading G at position 1 in the gRNA? Is the G/C content over 75%?

10  Off-target Count: The number of off-target sites the 5′ and 3′ gRNA may target.

## Data availability

The stand-alone application and code produced in this study are available in the following location/databases: Software: https://

www.laboratorychild.com/crispr-tape; Code: GitHub (https://github.com/LaboratoryChild/CRISPR-TAPE).

**Expanded View** for this article is available online.

## Acknowledgements
We would also like to thank Professor Mike Sternberg, and Dr Ellen M. McDonagh for critical reading of the manuscript, and beta-testers in the Child and Tate laboratories. This work was supported by grants BB/M011178/1 from the BBSRC (to H.J.B., E.W.T. and M.A.C.) and 202553/Z/16/Z from the Wellcome Trust & Royal Society (to M.A.C.).

## Author contributions
Conceptualization: MAC, HJB, DPA; Investigation: HJB, DPA; Formal analysis: MAC, HB; Visualization: MAC, HJB, DPA; Writing—original draft: MAC; Writing —review and editing: MAC, EWT, HJB, DPA; Supervision: MAC, EWT; Funding acquisition: MAC, EWT.

## Conflict of interest
The authors declare that they have no conflict of interest.

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
