## [Review Process File · Molecular Systems Biology]

CRISPR-TAPE: protein-centric CRISPR guide design for targeted proteome engineering

Daniel Anderson, Henry Benns, Edward Tate, and Matthew Child

DOI: [10.15252/msb.20209475](https://doi.org/10.15252/msb.20209475)

Corresponding author(s): *Matthew Child (m.child@imperial.ac.uk)*

Review Timeline:

First Submission Date:	24th Jan 20
Editorial Decision:	28th Jan 20
Second submission:	11th Mar 20
Editorial Decision:	14th Apr 20
Revision Received:	26th Apr 20
Editorial Decision:	30th Apr 20
Revision Received:	30th Apr 20
Accepted:	4th May 20

Editor: Maria Polychronidou

Transaction Report:

28th Jan 2020

Manuscript Number: MSB-20-9475

Thank you for submitting your manuscript titled "CRISPR-TAPE: a protein-centric CRISPR guide design tool for targeted proteome engineering" to Molecular Systems Biology.

I have now had the chance to read your manuscript and I regret to inform you that we that we have decided to not send it out for peer review.

In this study, you present CRISPR-TAPE, a protein-centric CRISPR guide RNA (gRNA) design algorithm for Targeted Proteome Engineering. We appreciate that you report that compared to gene-centric algorithms CRISPR-TAPE does not require post-hoc curation, simplifies gRNA outputs and reduces CPU times for gRNA identification. Given the focus of the manuscript on reporting a new methodology, we think that a Correspondence does not seem to be the most suitable format. It seems more appropriate to consider the study as a Method. While we acknowledge the speed and efficiency of the approach, we feel that in itself this improvement would not seem to provide the kind of decisive methodological advance that would be required for publication in Molecular Systems Biology. Even though we cannot offer to publish your manuscript in its current form, we recognize that CRISPR-TAPE could be a potentially relevant methodology. As such, we would not be opposed to considering editorially an extended manuscript, including a demonstration of the application(s) and advantages of CRISPR-TAPE for protein engineering e.g. for targeted protein modification or protein evolution.

I am very sorry to have to disappoint you on this occasion, but I hope that this early decision will allow you to decide how to proceed with your manuscript without undue delay.

Yours sincerely,

Maria Polychronidou, PhD
Senior Editor
Molecular Systems Biology

This revision was an invited resubmission and therefore contained no point by point response.

14th Apr 2020

Manuscript Number: MSB-20-9475R

Title: CRISPR-TAPE: protein-centric CRISPR guide design for targeted proteome engineering

Thank you again for submitting your work to Molecular Systems Biology. We have now heard back from the three referees who agreed to evaluate your study. Overall, the reviewers think that the presented protein-centric approach to gRNA design is potentially useful for future applications. They raise however a series of concerns, which we would ask you to address in a revision.

As you will see below, the issues raised by the reviewers are rather clear and I think that there is no need to repeat the points listed below. Both reviewers #1 and #2 point out that experimentally validating some of the predicted gRNAs would significantly enhance the conclusiveness and impact of the study. They do indicate however, that such experiments may be outside the scope of the work. As such, we think that experimental validations are not mandatory for the acceptance of the study. That said, we would not be opposed to the inclusion of such data in case you have it already at hand or are willing to produce it.

Reviewer #3, in their succinct report, raise issues regarding the novelty of the study and we would therefore ask you to better emphasize in the text the novelty and significance of the work.

On a more editorial level, we would ask you to address the following.

REFEREE REPORTS

Reviewer #1:

In this paper, Anderson, Benns et al. developed a protein-centric approach to CRISPR gRNA design that they call CRISPR-TAPE. While currently available design software is gene-centric and optimized for gene disruption, this protein-centric approach will be very powerful, when specific point mutations, insertions or deletions have to be made to a protein site-specifically, which requires the gRNA to target a specific position within the coding sequence of the gene. I agree with the authors that this will be especially useful, when all amino acids of a certain type are to be mutated in order to identify attachment sites e.g. for posttranslational modifications. While I am not an expert for the design of such algorithms, I downloaded the software and was easily able to reproduce the

experiment that is given in the manuscript. I highly appreciate that the software is available in a stand-alone format and does not require additional tools. One thing that would be nice, would be to verify the performance of the designed gRNAs on a selected test case. Nevertheless, as I expect that these experiments cannot be done in a timely manner in the current situation, I would support publication without this verification experiment. I, therefore, think that this manuscript is suitable for publication in Molecular Systems Biology after a few minor comments have been addressed.

- I think it would be good for the non-expert reader, to add a little bit of information on the workflow, how gene-editing with CRISPR is done. This should include, why the CRISPR cleavage positions at certain distances to the site of interest are needed in certain cases and what the problems with gRNA design are. Also there might be a sentence to state that for gene disruption the position in the gene is less important and that is why traditional tools are designed in the way they are. In this way, there would be a clearer path to why the software was designed in the way that it is.

- During my evaluation of the software the first attempt that I started from the Desktop did not work. There might be some issue with the long pathname or I might have made a copy-paste mistake. In the second try from a different location it worked nicely. In the failed case, the software just did not give any response. I guess some kind of error messages, bug reports or at least prompts that there was an error would be good to add. Especially considering that the user is encouraged to send back error reports.

- Figure 2 and S2 are very hard to read in the current size and resolution. I think it would be easier and better for the user to also have them as Excel tables (especially Figure S2)

- At the last page of the SI, the Citation of this paper is given as "TBC". I understand the reasoning for use in the program, but would delete this from the manuscript.

Reviewer #2:

Summary: Anderson et al. describe CRISPR-TAPE as a new tool that enables the rapid identification of gRNAs proximal to specific sites in coding sequencing. This tool represents a useful mechanism for gRNA prioritization for example for CRISPR-based mutagenesis screens. The key innovation of this study is that it takes a protein-centric approach to gRNA design, and as a result is uniquely suited to the production of point mutations in coding sequences. Another key result of this study is that it speeds up the production of guides, reduces computing time, and decreases requirements for manual curation.

Recommendation: This study represents an exciting tool for gRNA design with applications to chemoproteomics and chemical biology. However, several points should be addressed prior to consideration for publication in MSB.

- 1) Could gRNAs for base editors be incorporated into the algorithm? Given the low efficiency of homology directed repair (HDR), the use of base editors would likely be a more efficient strategy to generate point mutations. If incorporation into the package is not possible, discussion of the limitations of HDR for producing point mutations should be included in the text.
- 2) The authors should more clearly highlight the rationale and importance behind why they chose to prioritize proximity to nuclease cut sites. What other features do existing programs prioritize? Is this a novel feature? Has it been implemented in other algorithms?
- 3) CPU run time was used to assess algorithm efficiency in comparison to manual curation.

However, this algorithm was only compared to two other gRNA generating algorithms, EuPaGDT and CHOPCHOP. The authors commented how the difference in "consortia-funded data servers" make algorithm comparisons unequal. Are there other "stand-alone" packages/software/programs that they could benchmark their package against to provide a fairer comparison of CPU run time? What about other benchmarks, such as memory/power consumption, latency between user input and program execution?

4) Modification of Fig 1 to show both CRISPR-TAPE and a traditional gene-centric workflows would make the manuscript more accessible.

5) While it may be beyond the scope of this text, the manuscript would be significantly strengthened with the addition of experimental validation of some identified guides.

6) Figure S2 is extremely small and hard to interpret and should be modified for clarity.

Reviewer #3:

The authors present a protein-centric gRNA search tool which enables fast gRNA search to target specific regions or amino acids within a protein. The authors claims that the tool might be helpful to modify specific amino acid such as a catalytic residue, which means base editing of the sequence. But currently there are Crispr editors independent of double-strand breaks and homology-directed repair, which reduce the imporantance of this manuscript. Thus, the manuscript did not provide significant appovement to the field.

We would like to thank you and all three reviewers for taking the time to carefully consider our work, and for their specific reviews. We have addressed all the points raised, and as a result feel that the manuscript has been further strengthened.

The following details our point-by-point response to the specific reviewer comments for Anderson & Benns *et al.* For clarity of our responses, we have kept reviewer comments in black text, our direct responses are in blue text, and text additions to the main manuscript can be identified as red text, both here and in the main manuscript where they have been inserted.

Reviewer #1:

In this paper, Anderson, Benns et al. developed a protein-centric approach to CRISPR gRNA design that they call CRISPR-TAPE.

While currently available design software is gene-centric and optimized for gene disruption, this protein-centric approach will be very powerful, when specific point mutations, insertions or deletions have to be made to a protein site-specifically, which requires the gRNA to target a specific position within the coding sequence of the gene. I agree with the authors that this will be especially useful, when all amino acids of a certain type are to be mutated in order to identify attachment sites e.g. for posttranslational modifications. While I am not an expert for the design of such algorithms, I downloaded the software and was easily able to reproduce the experiment that is given in the manuscript. I highly appreciate that the software is available in a stand-alone format and does not require additional tools.

One thing that would be nice, would be to verify the performance of the designed gRNAs on a selected test case. Nevertheless, as I expect that these experiments cannot be done in a timely manner in the current situation, I would support publication without this verification experiment.

- We appreciate Review 1's understanding of the utility of the tool, as well as time issues in regard to validation of guides. We reiterate that while CRISPR-TAPE allows for the process of guide design to be initiated at the level of a protein target, the rules governing guide identification unchanged, and so guides are functionally equivalent to those generated by other tools (and present within standard guide outputs) but do not require subsequent extensive and time consuming manual curation to identify.

I, therefore, think that this manuscript is suitable for publication in Molecular Systems Biology after a few minor comments have been addressed.

- I think it would be good for the non-expert reader, to add a little bit of information on the workflow, how gene-editing with CRISPR is done. This should include, why the CRISPR cleavage positions at certain distances to the site of interest are needed in certain cases and what the problems with gRNA design are. Also there might be a sentence to state that for gene disruption the position in the gene is less important and that is why traditional tools are designed in the way they are. In this way, there would be a clearer path to why the software was designed in the way that it is.

- As the reviewer suggests, the following brief introduction to CRISPR has been added to **page 1, paragraph 1**:

“CRISPR technologies depend on the targeting of an RNA-guided endonuclease to a defined sequence location within the genome. This system has been harnessed for a variety of genome modification strategies including gene knockouts (through incorrect repair of breaks), and site-directed mutagenesis (though increased efficiency of homology-directed repair incorporating DNA templates at genomic regions close to the breaks). In most applications accurate targeting of the nuclease to the genomic locus takes priority over the specific nucleotide position of enzymatic activity. While engineering of the nuclease has driven diversification of the technologies that this system can support (Pickar-Oliver & Gersbach, 2019), the molecular rules governing nuclease

targeting remain the same; the genome address is encoded within a guide RNA sequence (gRNA), defined as a 20-nucleotide stretch of genomic DNA preceding a protospacer-adjacent motif (PAM)."

- And, we have expanded upon the text in the manuscript to provide additional detail and a clearer path towards the underlying rationale for the software design, with the following text added to added to **page 2, paragraph 2:**

"This is important as the increased efficiency of homology-directed repair (HDR) at double-strand breaks driven by the activity of the nuclease has a limited range. The efficiency of HDR decreases with increasing distance from the nuclease cut site, up to a maximum range of 30 nt (Paquet et al, 2016). If the distance of the mutation site of interest from the nuclease cut site is greater than 30 nt, there is no increase in the efficiency of HDR afforded by the use of the RNA-guided nuclease. This emphasizes the need to account for this when selecting gRNAs for directed mutagenesis strategies, where optimal HDR is essential."

- During my evaluation of the software the first attempt that I started from the Desktop did not work. There might be some issue with the long pathname or I might have made a copy-paste mistake. In the second try from a different location it worked nicely. In the failed case, the software just did not give any response. I guess some kind of error messages, bug reports or at least prompts that there was an error would be good to add. Especially considering that the user is encouraged to send back error reports.

- We appreciate the lack of an error box pop-up in certain use scenarios is not helpful to the user, and during the process of developing the tool we endeavoured to identify *all possible* error situations, but this is obviously a challenging task. We have added the following specific point to the troubleshooting section of the app-associated README, see **page 16, final bullet point of troubleshooting:**

"It is possible that in certain unanticipated user situations error-pops might not appear, indicating a new error that has not been previously identified. Please report as requested for bugs (see below)."

- We are continuing to write and deployed new versions of the script to ensure that once identified and reported, use error scenarios are recognized via a pop-up window. New versions have been deployed to ensure that in all situations of correct script launch but completion without guide generation, an error pop-up occurs.

- Figure 2 and S2 are very hard to read in the current size and resolution. I think it would be easier and better for the user to also have them as Excel tables (especially Figure S2)

- As requested, we have included higher resolution tables in Figure 2, and replaced Figure S2 with a Supplementary excel data table (now table EV1).

- At the last page of the SI, the Citation of this paper is given as "TBC". I understand the reasoning for use in the program, but would delete this from the manuscript.

- We apologise for this. The SI is a direct copy of the app-associated README, which has been live and available for community use since February alongside the tool. The citation will be retained within the downloadable README, but has been removed from the SI as requested.

Reviewer #2:

Summary: Anderson et al. describe CRISPR-TAPE as a new tool to that enables the rapid identification of gRNAs proximal to specific sites in coding sequencing. This tool represents a useful mechanism for gRNA prioritization for example for CRISPR-based mutagenesis screens. The key innovation of this study is that it takes a protein-centric approach to gRNA design, and as a result is uniquely suited to the production of point

mutations in coding sequences. Another key result of this study is that it speeds up the production of guides, reduces computing time, and decreases requirements for manual curation.

Recommendation: This study represents an exciting tool for gRNA design with applications to chemoproteomics and chemical biology. However, several points should be addressed prior to consideration for publication in MSB.

1) Could gRNAs for base editors be incorporated into the algorithm? Given the low efficiency of homology directed repair (HDR), the use of base editors would likely be a more efficient strategy to generate point mutations. If incorporation into the package is not possible, discussion of the limitations of HDR for producing point mutations should be included in the text.

- We appreciate the reviewer's comments and consideration about the future potential of the tool in relation to other members of the growing CRISPR-Cas tool-box. Traditional base editors (e.g. Cas9-cytidine/adenine deaminase fusions) are still reliant upon gRNAs for their targeting to a region of interest. As such, users of these base-editing tools would be able to use CRISPR-TAPE in its current form, and not require special modifications to the script. While the base-editing functions of these systems are constantly improving, at best the enzymatic process of base editing is still somewhat imprecise and takes place optimally between positions 4-8 within the gRNA sequence 5' of the PAM. Furthermore, the specific activity of the current base editors enable C-to-T or A-to-G base editing. As such, some amino acids switches remain inaccessible, e.g. threonine-serine (to name but one of many). HDR remains a useful strategy for tailored protein mutagenesis strategies. For that reason we have initially focused the tool on supporting HDR-driven mutagenesis strategies, which are more precise. For future versions of CRISPR-TAPE it will be interesting to see if we can include a strategy to encode and incorporate a base editing search function. We are similarly considering how to adapt the script to support Prime-editing strategies. While these updates would certainly increase the future utility of the tool, we have not yet established appropriate algorithms to enable them. To support researchers seeking to do this more rapidly than we are currently able, we have already made the CRISPR-TAPE base code freely available via Github (referenced in the main body text), and ensured that our source code has been written to support the modular integration of script expansions in the future.
- We have recognized this specific use case identified by the reviewer, and included the following text re. base editors on **page 4, paragraph 2**:

“This would support CRISPR base editing using Cas9-cytidine/adenine deaminase fusions where particular amino acids switches are enzymatically inaccessible, such as threonine to serine. CRISPR-TAPE is focused on protein-led gRNA design, with guide outputs being for suitable for all CRISPR systems (e.g. base editors (Rees & Liu, 2018)).”

- And also on **page 5, paragraph 2**:

“The code can also be expanded to include amino acid motif-based proteome engineering strategies, batch processing, and next generation genome editing tools such as Prime Editing (Anzalone et al, 2019).”

2) The authors should more clearly highlight the rationale and importance behind why they chose to prioritize proximity to nuclease cut sites.

- A similar point was raised by reviewer 1, and in line with our response to that point the following text has been added to **page 2, paragraph 2**:

“This is important as the increased efficiency of homology-directed repair (HDR) at double-strand breaks driven by the activity of the nuclease has a limited range. The efficiency of HDR decreases with increasing distance from the nuclease cut site, up to a maximum range of 30 nt (Paquet et al, 2016). If the distance of the mutation site of interest from the nuclease cut site is greater than 30 nt, there is no increase in the efficiency of HDR

afforded by the use of the RNA-guided nuclease. This emphasizes the need to account for this when selecting gRNAs for directed mutagenesis strategies, where optimal HDR is essential.”

What other features do existing programs prioritize? Is this a novel feature? Has it been implemented in other algorithms?

- The features of other programs are program-dependent and typically prioritize disruption of the gene. This identification of gRNAs that direct nucleases close to a particular residue or amino acid type is a novel feature unique to CRISPR-TAPE that has not been implemented for other algorithms. We have included the following statement to make this clearer in the text on **page 2, paragraph 3**:

“To our knowledge, these features are unique to CRISPR-TAPE.”

3) CPU run time was used to assess algorithm efficiency in comparison to manual curation. However, this algorithm was only compared to two other gRNA generating algorithms, EuPaGDT and CHOPCHOP. The authors commented how the difference in "consortia-funded data servers" make algorithm comparisons unequal. Are there other "stand-alone" packages/software/programs that they could benchmark their package against to provide a fairer comparison of CPU run time? What about other benchmarks, such as memory/power consumption, latency between user input and program execution?

- Regarding overall timesaving, we envisage that the primary advantage of the protein focus removal of the need for manual curation, and that while the reduced processing was apparent and notable, it is secondary to the time saving in relation to the removal of need for manual curation of gRNAs. As the reviewer rightly points out, many factors can feed in to this, but the case for manual curation is clearer – i.e. when applied for protein engineering, manual curation is required for traditional gRNA outputs, but not necessary for CRISPR-TAPE outputs.
- We intentionally only provided limited comparison to existing tools as after much deliberation we felt that benchmarking for different tools is challenging and inherently unfair, for example, which ones would we compare of the >20 gRNA search tools available? As the scripts underlying the many different tools are fundamentally different in terms of their computational approach, and as CRISPR-TAPE provides a search mechanism not accessible and therefore not directly comparable to other algorithms, we have not heavily focused on comparing the efficiency of our script in relation to others. Fundamentally, the improvement in search speed is a reflection of the restricted search space that results from targeting specific amino acids or amino acid types, and not the entire gene locus. We attempted to undertake what we considered to be the “fairest” comparison possible – we performed gRNA searches for an entire locus using our script (e.g. not selecting an amino acid or amino acid type, and instead deploying our script in a more “traditional” mode). We consider this to be the fairest comparison as the underlying scripts and computational transformations for both searches are then identical, with the only difference being the protein-focused element (which is the focus of the CRISPR-TAPE).
- We have also taken into consideration the points raised by the reviewer, and expanded the statement regarding consortia-funded servers on **page 4, paragraph 2**:

“This difference could be due to a number of factors including the processing power of the consortia-funded data servers that support these community tools. Moreover, user dependent factors such as memory/power consumption, and latency times between user input and program execution makes direct comparison of processing speeds for different algorithms inherently unequal. Fundamentally, the improvement in computation processing speed is a reflection of the restricted search space that results from targeting specific amino acids or amino acid types, and not the entire gene locus.”

4) Modification of Fig 1 to show both CRISPR-TAPE and a traditional gene-centric workflows would make the manuscript more accessible.

- As suggested, we have updated figure 1 to include the gene-centric approach alongside the CRISPR-TAPE workflow. To accommodate this change we have shifted Figure 1b. (the lower resolution screen shot of the custom GUI) to the supplementary information.
- 5) While it may be beyond the scope of this text, the manuscript would be significantly strengthened with the addition of experimental validation of some identified guides.
- We are currently unable to undertake these experiments, but would like to stress that while CRISPR-TAPE allows for the process of guide design to be initiated at the level of a protein target, the rules governing guide identification and behaviour are fundamentally unchanged, and so the basic functionality of CRISPR-TAPE-generated guides are directly equivalent to those generated by other tools (and present within standard guide outputs) but do not require subsequent extensive and time consuming manual curation.
- 6) Figure S2 is extremely small and hard to interpret and should be modified for clarity.
- As requested, to improve clarity this has been replaced by a supplementary excel data table EV1.

Reviewer #3:

The authors present a protein-centric gRNA search tool which enables fast gRNA search to target specific regions or amino acids within a protein. The authors claims that the tool might be helpful to modify specific amino acid such as a catalytic residue, which means base editing of the sequence.

But currently there are Crispr editors independent of double-strand breaks and homology-directed repair, which reduce the imporantance of this manuscript.

Thus, the manuscript did not provide significant appovement to the field.

- We appreciate the reviewer's careful consideration of the manuscript, and consideration of the relevance of CRISPR-TAPE in relation to existing gRNA design tools. While tools for gRNA design are widely available, to our knowledge no existing tools allows for gRNA design to be undertaken from the level of the protein coding sequence. i.e. to make experimental design decisions (e.g. amino acids or amino acid types to target), and have protein-focused gRNAs provided in relation to the nucleotide position of the nuclease cut site from targeted protein feature of interest. Taking into the consideration the novelty of this aspect of our algorithm, we feel that our tool will be valuable for the field, and provide substantial time saving in relation to the normal manual curation required to filter and identify gRNAs outputted from traditional tools that target a particular amino acid of interest within the protein, and do not simply direct the CRISPR-associated nuclease to a non-specified region of the locus in order to introduce double strand breaks and the possibility of incorrect repair leading to a gene knockout. As described in our response to reviewers 1 and 2, we have more strongly stressed the novelty of the tool.
- In relation to other members of the growing CRISPR-Cas toolbox and repeating an earlier response, traditional base editors (e.g. Cas9-cytidine/adenine deaminase fusions) are still reliant upon gRNAs for their targeting to a region of interest. As such, users of these base-editing tools would be able to use CRISPR-TAPE in its current form, and not require special modifications to the script. While the base-editing functions of these systems are constantly improving, at best the enzymatic process of base editing is still somewhat imprecise and takes place optimally between positions 4-8 within the gRNA sequence 5' of the PAM. Furthermore, the specific activity of the current base editors are restricted to C-to-T or A-to-G base editing. As such, some amino acids switches remain inaccessible, e.g. threonine-serine (to name but one of many). HDR remains a useful strategy for tailored protein mutagenesis strategies. For that reason we have initially focused the tool on supporting HDR-driven mutagenesis strategies, which are more precise. For future versions of CRISPR-TAPE it will be interesting to see if we can include a strategy to encode and

incorporate a base editing search function. We are similarly considering how to adapt the script to support Prime-editing strategies. While these updates would certainly increase the future utility of the tool, we have not yet established appropriate algorithms to enable them. To support researchers seeking to do this more rapidly, we have already made the CRISPR-TAPE base code freely available via Github (referenced in the main body text), and ensured that our base code has been written to support the modular integration of script expansions in the future. This means that the underlying algorithm has the potential to support uses tangential to the original guide search function outlined in this manuscript.

Thank you for sending us your revised manuscript. We think that the performed revisions have satisfactorily addressed the issues raised by the reviewers. As such, I am glad to inform you that your manuscript is now suitable for publication, pending some minor editorial issues listed below.

We would ask you to address the following in a minor revision.

The Authors have made the requested editorial changes.

4th May 2020

Manuscript number: MSB-20-9475RRR

Title: CRISPR-TAPE: protein-centric CRISPR guide design for targeted proteome engineering

Thank you again for sending us your revised manuscript. We are now satisfied with the modifications made and I am pleased to inform you that your paper has been accepted for publication.

Corresponding Author Name: Matthew Child

Manuscript Number: MSB-20-9475R